# Efficient sepsis detection using deep learning and residual convolutional networks

Ahmed S. Almasoud[1], Ghada Moh Samir Elhessewi[2], Munya A. Arasi[3], Abdulsamad Ebrahim Yahya[4], Menwa Alshammeri[5], Donia Badawood[6], Faisal Mohammed Nafie[7] and Mohammed Assiri[8]

[1] Department of Information Systems, College of Computer and Information Sciences, Prince Sultan University, Saudi Arabia
[2] Department of Health Sciences, College of Health and Rehabilitation Sciences, Princess Nourah bint Abdulrahman University, Saudi Arabia
[3] Department of Computer Science, Applied College at RijalAlmaa, King Khalid University, Saudi Arabia
[4] Department of Information Technology, College of Computing and Information Technology, Northern Border University, Arar, Saudi Arabia
[5] Department of Computer Science, College of Computer and Information Sciences, Jouf University, Saudi Arabia
[6] Department of Data Science, School of Computers, Umm Al-Qura University, Saudi Arabia
[7] Department of Natural and Applied Sciences, Community College, Majmaah University, Al Majma'ah, Saudi Arabia
[8] Department of Computer Science, College of Computer Engineering and Sciences Prince Sattam Bin Abdulaziz University, Al-Kharj, Saudi Arabia



Corresponding author
Abdulsamad Ebrahim Yahya,
Abdulsamad.qasem@nbu.edu.sa

## ABSTRACT

Sepsis is a life-threatening complication caused by infection that leads to extensive tissue damage. If not treated promptly, it can become fatal. Early identification and diagnosis of sepsis are critical to improving patient outcomes. Although recent technological advancements have aided sepsis detection, challenges remain in timely diagnosis using standard clinical practices. In this article, we present a new deep learning model to detect the occurrence of sepsis and the African vulture optimization algorithm (AVOA) to enhance the model performance. The system comprises four crucial steps: First, the enhanced convolutional learning framework (ECLF) with atrous convolutional and multi-level strategies that aim to learn high-level features from the nonlinear mapping of the medical data. Second is the spatio-channel attention network (SCAN), which has a neural architecture designed to focus on significant regions, such as spatial and channel regions, but not restricted to them. Third is the hierarchical dilated convolutional block (HDCB), which utilises a stacked dilated deep convolutional architecture for spatial feature context retrieval. Last is the residual path convolutional chain (RPCC), which uses a multi-residual convolutional approach for feature propagation, preserving important information. The sepsis detection model we bring forth involves many components, as mentioned above, and thus achieves a higher accuracy for timely intervention during sepsis. The combination of AVOA into the model ensures that it is robust and easily transferable, delivering high performance for adaptation to complicated structures inside medical datasets. The proposed model was evaluated on a clinical dataset and achieved outstanding performance, with an accuracy of 99.4%, precision of 98%, recall of 99.2%, F1-score of 99.0%, and an area under the curve (AUC) of 0.998. These results demonstrate the model's superior ability to detect sepsis accurately and reliably,

**Peer**J Computer Science

> outperforming traditional clinical scoring methods and conventional machine learning approaches.

# INTRODUCTION

Sepsis is a critical health condition resulting from an abnormal response to infection. It ranks as the sixth leading cause of death worldwide, responsible for more than 11 million fatalities each year. Early fluid therapy, antibiotics, and source control management are essential to improve survival rates. However, diagnosing sepsis is challenging due to its complex presentation and rapidly changing symptoms. Current diagnostic methods often lack sensitivity and fail to predict outcomes accurately, making the detection process time-consuming and unreliable (*Rudd et al., 2020*). Since septic shock is the most severe form of sepsis, the management of sepsis must begin early through fluid and antibiotic therapy as well as source control. Unfortunately, sepsis is more complex than it seems, owing to its very disparate characteristics, and many of the current approaches aimed at determining whether an individual has sepsis are either inadequately sensitive or inaccurately predict the sepsis outcome, rendering the sepsis detection process time-consuming. As a result of this rapid adoption, predictive algorithms have been developed and utilised to improve the likelihood of finding sepsis at the right time (*Henry et al., 2015*). Currently, the approach recommended for diagnosis of neonatal sepsis involves bacteriologic examination of blood cultures performed before commencing treatment with antibiotics. Nevertheless, sepsis, unable to be treated in time, can have very non-defining symptoms and lead to organ failure within no time. That is why, while the cultures are pending, patients are usually given empirical treatment with ampicillin, gentamicin, or cefotaxime (*Hussaini, 2021*).

Even though there are clinical scoring systems like Sequential Organ Failure Assessment (SOFA) and Simplified Acute Physiology Score II (SAPS II), such methods have many shortcomings. Their dependence upon empirical findings and antiquated markers lowers their applicability in contemporary medicine, particularly considering the accelerated course of sepsis that hinders the development of data relative to the patient as opposed to chronic diseases where such illness develops (*Hu et al., 2022*). Hence, estimating the mortality rate in cases of sepsis continues to be complex. To appreciate these limitations, using machine learning (ML) methods to improve estimating sepsis outcomes has become widespread (*Hu et al., 2022*). Moreover, since the definition of sepsis is poorly defined in the literature, which makes comparing various predictive pre-model and post-model surveys even more complex, the lack of well-defined, cross-validation enabled, multi-centre databases has made it impossible to validate sepsis models externally (*Wong et al., 2021*). Sepsis is still a leading cause of death, where cases remain unabated, resulting in death rates

that are over 30%. Long-term complications such as physical disability, cognitive disability, and psychological disability are common among survivors of sepsis. A series of disabilities powerfully puts a strain on the healthcare systems. The Surviving Sepsis Campaign states that one year after a discharge, 33% of survivors display signs of cognitive impairment, 43% are not able to perform some tasks, and 27% have post traumatic stress disorder (PTSD) (*Iwashyna et al., 2010*).

Immune system malfunction is widely seen in sepsis. The infection affects the immune system with such force that the organism cannot fight off invading microorganisms. Although some antibiotics work on sepsis with positive results, the shipment of fighting the infection can lead to an attack of resistance. There are several issues with early diagnosis, but the recent paradigm shift in the concept of the disease provides sufficient information for its explanation and selection of effective antibiotics (*Singer et al., 2016*). While space-limited clinical notes filled with typos/abbreviations/free text are complicated, structured clinical information such as age, lab tests, and vital signs is pretty straightforward. Helpful information is retrieved from this unstructured data by employing natural language processing techniques; these often require clinicians' intervention (*Spasic & Nenadic, 2020*). Considering the latest technology trends, this article proposes improving sepsis detection using artificial intelligence (AI) algorithms, which can find more relevant sepsis indicators and risk factors from large clinical datasets, including echocardiograms, electrodes, and many others. Although AI also faces this problem, it is solved through the fast analysis of vast datasets, which minimises errors related to human interpretation and estimate while enhancing the speed and reliability of dosages (*Giordano et al., 2021*).

The widespread adoption of electronic health records (EHRs) has made it feasible to incorporate machine learning and data mining techniques into the fight against sepsis. Unfortunately, tools like quick Sepsis-related Organ Failure Assessment (QSOFA), Modified Early Warning Score (MEWS), National Early Warning Score (NEWS), and Systemic Inflammatory Response Syndrome (SIRS) also have problems. They do not predict sepsis's occurrence, making it difficult for early treatment (*Bone et al., 1992*). The treatment of sepsis is vital, mainly because the more extended treatment is delayed, the chances of death increase by 4 to 8% each hour (*Parlato et al., 2018*). There have been approaches targeting different possible biomarkers to diagnose sepsis, but none have proven practical in clinical settings (*Parlato et al., 2018*). Due to the outstanding complexity of sepsis, which is dynamic and includes both the infection and the host response, accurate prediction is difficult (*Iskander et al., 2013*).

Sepsis continues to be an important health issue threatening people around the world, causing more than 11 million deaths each year–surpassing many other infectious diseases in mortality. Even with technological advancements, timely detection is still out of reach because of the disease's intricate progression and vague early symptoms. SOFA and SAPS II scores are more traditional diagnostic methods that heavily depend on clinical benchmarks and slow lab results, hitting a brick wall when trying to support immediate action. In this research, a new framework is proposed, guided by AI technology that incorporates real-time clinical data through deep learning constructs for precise and early

sepsis identification. With this model, real-life problems where healthcare professionals can execute treatment hours in advance are the focus, saving thousands of people, which can be considered the most valuable feature. ICU and in-hospital settings are not the only areas where this research can be helpful. It can significantly change the odds of patient survival, lessen long-term disabilities among patients, and lessen the burden on healthcare systems globally. Delayed treatment is known to escalate mortality rates by 8% every hour, so integrating such predictive models with electronic health record's label systems will alter clinical thinking for the better—lowering mortality rates and enhancing the quality of life for patients around the globe.

The significant contributions of this article are,

1. This work proposes a novel hybrid deep learning model for early sepsis detection, combining convolutional neural networks (CNN) with spatio-channel attention mechanisms and residual path convolutional chains.
2. This model is claimed to overcome the drawbacks of conventional clinical scoring systems by exploring multidimensional clinical data, such as structured patient data, clinical notes, and unstructured data.
3. Moreover, the article provides the African vulture optimization algorithm (AVOA) approach for model tuning and performance enhancement efforts. The model aims to provide a reasonable prediction of the time of sepsis development and, thus, improve patient treatment by preventing sepsis.

This study has taken a quantitative approach using deep learning algorithms and analytics on clinical datasets, although the application of mixed-method strategies is recognised. For the case of sepsis detection, a mixed approach—blending machine analysis with clinical expert narratives—can be helpful for issues related to explainability, model output confidence, deployment, and acceptance by healthcare providers. However, because this effort's scope is technical and focused on the design, optimization, and performance evaluation of clinical models with structured clinical datasets, qualitative elements such as clinician interviews, surveys, or ethnographic studies were not included in this work. This blended approach may help future research explore gaps focused on practical barriers and user-centred design evaluation in clinical workflows. SCr, SOFA, SIRS, and MEWS as clinical scoring systems give a relatively weak response both temporally and in sensitivity. They also rely heavily on the clinical input and/or laboratory results, which are usually slow and unmanned. Given the nature of these systems, they become less valuable in critical situations where prompt action is needed. In addition, many current machine learning approaches also lack the ability to deal robustly with incomplete and heterogeneous patient data, or they deal with the explicit temporal patterns in clinical data observations.

To address these challenges, the model proposes a hybrid deep learning architecture that includes spatio-channel attention mechanisms, hierarchical dilated convolutional blocks, and residual path convolutional chains. This configuration permits precise multi-scale feature extraction and guided attention learning, improving predictability

while ensuring interpretability. Furthermore, model accuracy and generalization are optimized using the AVOA for hyperparameter tuning, advancing model performance. Thus, this study builds an advanced yet clinically useful model designed to enable reliable and prompt detection of sepsis in operational healthcare settings, closing gaps in computation and clinical focus within the literature.

The article is organised as follows: "Related Works" surveys the literature on sepsis detection based on traditional approaches and those relying on machine learning. "Methods and Materials" presents the details of the hybrid deep learning architecture, including architecture, preprocessing and optimisation. "Result and Discussion" presents the cases used in the experiments and evaluates the results, including the evaluation with baseline models, ablation studies, and exploring hyperparameters settings. "Conclusion" provides the article's conclusions, indicating the prospects offered by the new model for improved sepsis detection and further research development.

## RELATED WORKS

Using critical health indicators and laboratory results, usually available within several hours of an emergency admission, *Faisal et al. (2018)* created a logistic regression model (CARS) to forecast the likelihood of sepsis. An ML model for sepsis clinical decision assistance in the ED was constructed by *Horng et al. (2017)* using a linear support vector machine. They showed that incremental improvements include free text input, vital signs, and demographic data (*Horng et al., 2017*). This methodology relies on the rapid collection of patients' medical parameters. Attempting to gather patient data from a more extended period, the second model prediction of long-term sepsis explored the prediction. This study considered 33 patients with an average age of 49. Of these, 19% were female, and the remaining patients were male. A more intricate framework and suitable medical parameters are necessary to detect long-term sepsis. A technique that aids in the early identification of various degrees of sepsis was created by *Burdick et al. (2017)*.

The exact location also provided the data for a distinct cohort used for temporal validation. All visits to the emergency department (ED) between March 1, 2018, and August 31, 2018, were included in the cohort, not only inpatient admissions. Everything from the variables used to define the outcomes to the criteria for inclusion and exclusion remained constant. Interactions that started in the emergency department but did not lead to inpatient admission were included in the temporal validation cohort, in contrast to the model development cohort (*Hochreiter & Schmidhuber, 1997*). The report emphasises that sepsis is a significant problem in medicine all around the world, causing millions of cases and deaths annually, especially in susceptible groups like youngsters. This highlights the critical need for better sepsis diagnosis tools (*Daothong, Jampa-ngern & Senavongse, 2024*). The intensive care unit (ICU) makes sepsis diagnosis even more challenging because many patients there have other illnesses that present with identical laboratory and physiological changes as sepsis (*Pierrakos et al., 2020*).

ICU outcomes and sepsis prediction have long piqued the curiosity of medical professionals. Computer scientists are becoming more interested in finding practical answers to this challenge because of its importance. Modern computers' improved

processing power, along with the encouraging results seen by AI and deep learning techniques in various contexts, has cleared the way for more efficient and effective methods of studying and analysing sepsis (*Vellido et al., 2018*). If septic patients can be identified early and given the right drugs when they need them, their prognosis and chances of survival will be much improved (*Marik & Farkas, 2018*). Because of the growing complexity and possible impact, it is crucial to comprehend the implementation policy layer, which records the operational restrictions, reaction protocols, and clinical workflow. Specifically, the area under the receiver operating characteristic curve, which is one of the most popular ML evaluation methods, frequently disregards the impact of this policy layer on model performance (*Reyna, Nsoesie & Clifford, 2022*).

A significant contributor to healthcare costs, sepsis is a leading cause of hospital and ICU admissions, as well as morbidity and mortality. To prevent additional organ dysfunction and restore a normal circulating blood volume, intravenous fluids and/or vasopressors are essential for sepsis treatment. The problem is that human doctors have difficulty figuring out when and how much of these therapies to administer (*Van der Ven et al., 2022*). The targeted real-time early warning score (TREWScore) was suggested by *Henry et al. (2015)* as a means of predicting septic shock using the Cox-proportional hazard model. When specific clinical characteristics change over time, this support system will sound an alarm to alert the patient to the possibility of septic shock. The suggested model attained a discrimination ability of 0.83. This analysis does not address the likelihood of mortality and fails to account for the impact of preexisting illness conditions. When it comes to sepsis, the majority of expert systems treat clinical aspects separately. However, physiological variables are probably not independent (*Henry et al., 2015*). Significant constraints, such as a lack of validation and variations in the definition of sepsis, limit their generalisability and prevent any of these models from being widely employed in the clinic, even if some demonstrate excellent discriminative power (*Kang et al., 2008*).

One of the biggest problems with studying sepsis in emergency departments is the absence of trustworthy labels. Due to the absence of a universally accepted standard, sepsis modelling labels are currently generated using diagnostic tools and claims-based procedures. However, it is well-known that these tagging approaches are not ideal (*de Hond et al., 2022*). While this approach did a decent job of making predictions, it could not detect changes that occurred over time since it did not consider patients' temporal data. Further increases in predicting performance were likely limited by the model's simplicity, which hindered its capacity to detect complicated data patterns. One model, *Gong et al.*'s *(2022)*, integrated time-series data with four vital signs to predict near-term mortality risks in sepsis patients. However, this model only considered a few variables. A total of ninety-two ICU patients had their blood tested for the start of sepsis using real-time polymerase chain reaction (RT-PCR) expression and genetic network analysis. In this investigation, 83.09% of cases could be predicted 1 to 4 days before the clinical diagnosis, with a specificity of 80.20% and a sensitivity of 91.43% (*Lukaszewski et al., 2008*). *Raghu et al. (2017)* utilised a Duelling Double-Deep Q Network in continuous space to develop medical treatment plans for sepsis. This method expanded sepsis treatment to a continuous space using a vector

representation of constant conditions (*Raghu et al., 2017*). *Sun et al. (2021)* used a hybrid approach that combined supervised learning and reinforcement learning using the deep deterministic policy gradient (DDPG) method to create strategies in a continuous value space.

The study to trace the interactions of berberine with inflamed molecular targets, proteins, and pathways of interest (*Wang et al., 2024*; *Sun et al., 2025*). This allows simultaneous detection of both pathogens in one assay without separate reactions (*Zhang et al., 2025*).

Using data from electronic health records (EHRs), (*Lauritsen et al., 2020*) created a deep-learning model to identify sepsis in its early stages. They drew on electronic health record data collected retrospectively from various Danish hospitals, encompassing ICUs and other departments over 7 years. Time stamps from things like lab tests, notes, prescriptions, and more were included in the data. They used sparse vectors to represent each occurrence. They built a convolutional neural network long short-term memory (CNN-LSTM) model consisting of CNN layers to capture temporal dependencies and extract features from event sequences. They pitted it against multilayer perceptron (MLP) and gradient boosting on vital signs (GB-Vital), two baseline models. At 3 h before the start of sepsis, the CNN-LSTM model outperformed the baselines with an AUROC of 0.856. Even before the beginning, 24 h before, the area under the receiver operating characteristic (AUROC) was 0.756. To measure the model's value across various forecasts made throughout a patient's hospital stay, they suggested a novel "sequence evaluation" method. They also checked the algorithm's usefulness by counting the number of hopeful forecasts already receiving antibiotics or having blood cultures taken to determine how much sooner the model could have assisted. To summarise, they created a sequential deep learning model for early sepsis diagnosis that was more effective than previous models and suggested new ways to evaluate its clinical usefulness. The concept has the potential to be applied to several departments within the hospital (*Lauritsen et al., 2020*).

## METHODS AND MATERIALS

This research aims to develop a method for enabling early and timely detection of sepsis, which is a significant clinical challenge because it increases the risk of death by 4% to 8% with every hour treatment is delayed. Routinely used clinical scoring systems disregard aiding the patient until sepsis is well underway, and then, treatment options are minimal and outcomes are grim. Unlike the proposed model, which endeavours to predict sepsis at the earliest possible stage using real-time analyses of continuously collected clinical data, including vital signs, lab results, and demographic details. The incorporation of sophisticated deep learning components such as the spatio-channel attention network (SCAN) and the hierarchical dilated convolutional block (HDCB) enables the model to detect the most subtle early-stage physiological alterations effectively, ensuring it is fit for use in clinical settings that require immediate action. Moreover, applying the African vulture optimisation algorithm (AVOA) further augments the model's accuracy and generalisation strength across diverse clinical settings and patient populations.

## Data collection

The Kaggle dataset (*Hussaini, 2021*) contains clinical data from patients diagnosed with sepsis, including vital signs, laboratory results, demographics, and other relevant clinical information. To prepare the data for models developed with deep learning, it is pre-processed to deal with missing values, normalise numerical parameters, and encode variable categories.

## Experimental setup

Python software was used for the experiments in a system with an Intel(R) Core(TM) i5-4590S CPU, 1TB HDD and 8GB of RAM memory.

## Dataset access

The dataset can be accessed at the following URL: https://www.kaggle.com/datasets/salikhussaini49/prediction-of-sepsis.

The dataset includes 44 clinical features, grouped as three vital signs, lab results, and demographic information.

Vital signs encompass measurements of heart rate (HR), oxygen saturation (O2Sat), body temperature (Temp), blood pressure (systolic SBP, mean MAP, and diastolic DBP), respiration rate (Resp), and EtCO2. Laboratory diagnosis encompasses white blood cell count, platelet count, lactate, bilirubin, creatinine, glucose, and other hormones related to organ function and infection. Demographic data contain the patient's age and gender and the time of hospitalization.

Each patient's information is carefully arranged as a series of time-stamped events, illustrating the evolution of clinical conditions through an evolving timeline. The target variable, SepsisLabel, marks the presence or absence of sepsis at each timestamp, allowing us to build models capable of predicting the onset of sepsis using past clinical data. The ordered characteristic of the dataset allows for the creation of predictive models that can identify the onset of sepsis in advance, given the urgency of the required clinical action.

## Assessment metrics

The evaluation metrics used to assess the proposed model—accuracy, precision, recall, F1, and AUC, classify its early sepsis detection capabilities as exceptional. Having an accuracy of 99.4%, the model also indicates the power of classifying the majority of cases correctly. At 98.7%, precision alone assures the model's reliability in correctly labeling actual cases of sepsis and lowering false positives. Recall value of 99.2% demonstrates the model's ability of capturing nearly all true sepsis cases, something that is important in clinical practice in order to not miss any diagnoses. The AUC of 0.998 which reflects the model's ability to discriminate sepsis from non-sepsis cases also validates the model's performance towards other metrics such as F1 and Recall. These metrics yield evidence of the model's reliability, generalizability, and applicability in deployment in healthcare for early sepsis detection, a life-critical task.

### Data preprocessing

Preprocessing is critical to ensure that the raw medical data is suitable for feeding into deep learning models. The preprocessing phase includes several key operations: handling missing data, normalisation, feature encoding, and outlier detection. Each step transforms the data into a consistent format and scale, allowing the deep learning model to learn from it effectively. Missing values are common in medical datasets, and handling them appropriately is crucial for the quality of the model. Missing values are filled in using statistical methods such as mean, median, or mode imputation. For numerical features with missing values, the missing entry is replaced using the mean of the available values, excluding the missing entry. In some cases, advanced techniques like multiple imputation by chained equations (MICE) or k-nearest neighbours (KNN) imputation can be employed if the missingness is not random. Medical data can span different ranges, with some features having significantly larger scales than others (*e.g.*, heart rate *vs.* body temperature). Normalising and scaling the data ensures all features are treated equally in the model's learning process. The Min-Max Scaling method usually scales the data to a specified range $[0, 1]$.

The Z-score normalization (standardization) method standardises the feature with a mean of 0 and a standard deviation of 1. Categorical variables were converted into numerical representations using one-hot encoding to ensure compatibility with the deep learning model. Numerical features were standardised using Z-score normalization to maintain consistent scaling across features. A common approach is the Z-score method, where data points that deviate more than a specified threshold from the mean are considered outliers.

Deviation. If $|Z_i| > 3$, the data point is flagged as an outlier and can be removed or adjusted. To further improve the robustness of the model, data augmentation techniques, such as noise injection or random cropping, may be applied, especially for limited datasets. This can help the model generalize unseen data better and prevent overfitting. By applying these preprocessing techniques, the dataset is transformed into a format that can be efficiently used for training deep learning models, ensuring better performance in the subsequent detection of sepsis.

$$x_i = \frac{1}{n}\sum_{i=1}^{n} x_i \text{ if } x_i \text{ is missing} \qquad (1)$$

where $\frac{1}{n}\sum_{i=1}^{n} x_i$ is the mean of the feature across the dataset. For time-series data, missing values are filled based on the previous or next observed values:

$$x_i = x_{i-1} \text{ or } x_i = x_{i+1} \text{ if } x_i \text{ is missing} \qquad (2)$$

In some cases, advanced techniques like MICE or k-nearest neighbours (KNN) imputation can be employed if the missingness is not random. Medical data can span different ranges, with some features having significantly larger scales than others (*e.g.*, heart rate *vs.* body temperature). Normalising and scaling the data ensures all features are treated equally in the model's learning process. The Min-Max Scaling method usually

scales the data to a specified range [0, 1]. For a feature $x_i$, the transformed value $x_i'$ is calculated as:

$$x_i' = \frac{x_i - min(x)}{max(x) - min(x)} \tag{3}$$

where $min(x)$ and $max(x)$ are the minimum and maximum values of the feature across all samples. The Z-score normalization (standardization) method standardises the feature with a mean of 0 and a standard deviation of 1. Categorical variables were converted into numerical representations using one-hot encoding to ensure compatibility with the deep learning model. Numerical features were standardised using Z-score normalization to maintain consistent scaling across features.

The transformation for each feature $x_i$ is given by:

$$x_i' = \frac{x_i - \mu_x}{\sigma_x} \tag{4}$$

where $\mu_x$ is the mean of the feature and $\sigma_x$ is the standard deviation of the feature. Standardisation is generally preferred when the model requires features to be on the same scale, especially for algorithms like gradient-based optimisation, which are sensitive to feature scaling. Categorical variables in the dataset, such as gender or ethnicity, need to be transformed into numerical values that the deep learning model can understand. Common methods include:

In One-Hot Encoding, each category is represented by a binary vector. For example, a categorical feature with three possible values (*e.g.*, "Male," "Female," "Other") would be encoded as:

$$Male \rightarrow [1, 0, 0], Female \rightarrow [0, 1, 0], Other \rightarrow [0, 0, 1] \tag{5}$$

This method ensures that the model does not assume ordinal relationships between categories. In Label Encoding, each category is assigned an integer value. For example, the categories "Male," "Female," and "Other" could be encoded as:

$$Male \rightarrow 0, Female \rightarrow 1, Other \rightarrow 2 \tag{6}$$

Label encoding is often used when the categorical feature has an inherent order. Still, it is not recommended for features with no ordinal relationship, as the model may misinterpret the encoding. Outliers can significantly distort training, especially in sensitive models like deep learning. Outlier detection and removal techniques help mitigate this issue. A common approach is the Z-score method, where data points that deviate more than a specified threshold from the mean are considered outliers. The Z-score for a feature $x_i$ is computed as:

$$Z_i = \frac{x_i - \mu_x}{\sigma_x} \tag{7}$$

where $\mu_x$ is the mean and $\sigma_x$ is the standard deviation. If $|Z_i| > 3$, the data point is flagged as an outlier and can be removed or adjusted. To further improve the robustness of the model, data augmentation techniques, such as noise injection or random cropping,

may be applied, especially for limited datasets. This can help the model generalise unseen data better and prevent overfitting. By applying these preprocessing techniques, the dataset is transformed into a format that can be efficiently used for training deep learning models, ensuring better performance in the subsequent detection of sepsis.

## Experimental setup

The experimental design for this work involved leveraging the Kaggle Sepsis Prediction dataset, which has 1,552,210 records from 40,336 unique patients. The target variable SepsisLabel marks whether sepsis is present or not, and has a notable class imbalance: 1,524,294 samples flagged as non-sepsis and 27,916 flagged as sepsis. This dataset is nearly representative of actual clinical situations, where early identification of sepsis is difficult because it is infrequently encountered. As a preparatory step towards model training, mean imputation was applied for handling missing values for numerical features, along with forward filling for time-series data where applicable. Categorical variables were transformed into one-hot variables. Numerical features were standardized using Z-score normalization to ensure uniform feature scaling. Using Z-score outliers detection, outliers were capped at three standard deviations. The dataset was split into three disjoint subsets for a patient-wise split to prevent data leakage: 80% for training (1,241,768 samples), 10% for validation (155,221 samples), and 10% for testing (155,221 samples). This ensures that all records from a single patient are not present in multiple subsets. The suggested architecture integrates state-of-the-art components of deep learning, namely enhanced convolutional learning framework (ECLF), spatio-channel attention network (SCAN), hierarchical dilated convolutional block (HDCB) and residual path convolutional chain (RPCC ). Model training was carried out on the Adam optimizer with a starting learning rate of 0.001, 128 batch size, and regularization set to a 0.5 dropout rate. The network was trained with ReLU activation and overfitting mitigated by early stopping based on validation loss, capping training at 100 epochs. The model's performance was assessed on multi-dimensional accuracy, precision, recall, F1-score, area under the ROC curve (AUC), specificity, FPR and FNR. All experiments were implemented on a Python 3.8 environment with TensorFlow and Keras libraries, running on a system with NVIDIA GPU and sufficient memory to handle the computations.

## Proposed methodology

The proposed model structure intends to capture more complex patterns and spatial-temporal dependencies in medical data for efficient sepsis detection. The architecture includes four broad components: ECLF, SCAN, HDCB and RPCC. Each module uses advanced techniques involving convolutional layers, attention mechanisms, and residual connections to improve the model's performance and facilitate feature extraction and model interpretability. Figure 1 shows the architecture of the Proposed methodology with ECLF, SCAN, HDCB and RPCC.

The enhanced convolutional learning framework, spatial-channel attention network, hierarchical dilated convolutional block, and residual path convolutional chain collectively form the advanced components of the deep learning architecture for the purposed sepsis

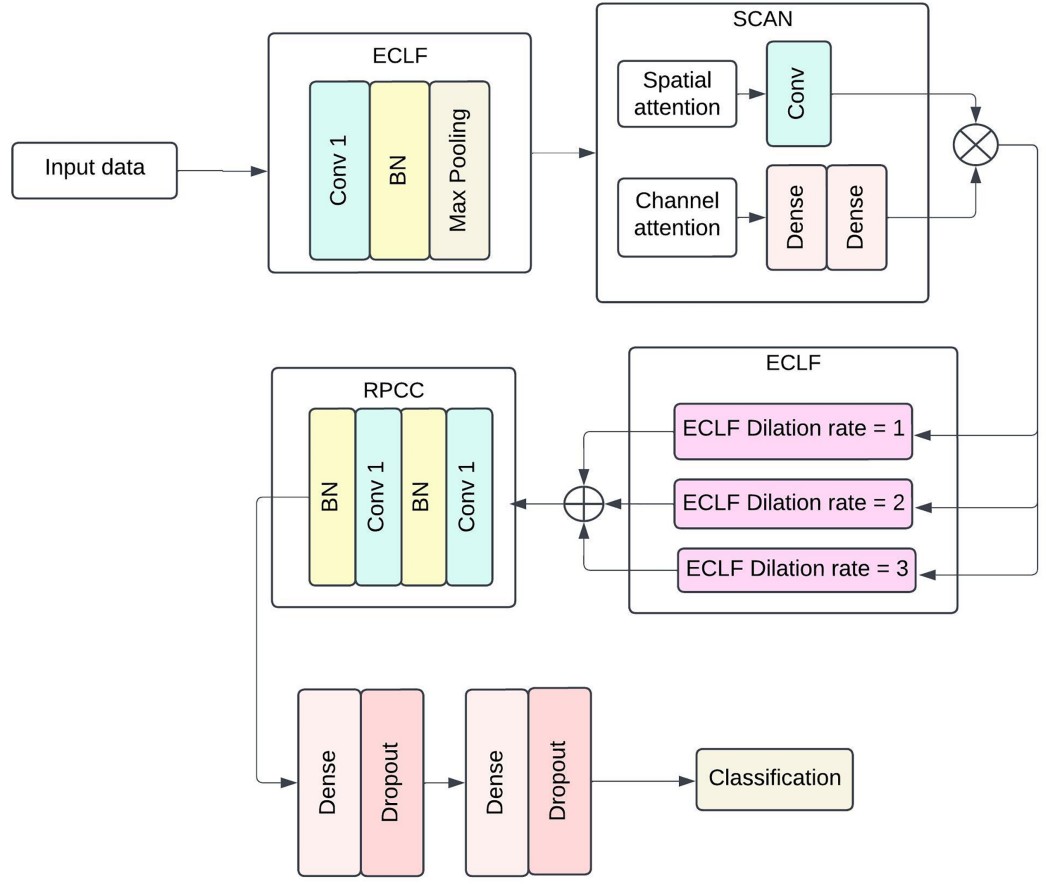

**Figure 1** **The architecture of proposed methodology with Enhanced Convolutional Learning Framework (ECLF), Spatio-Channel Attention Network (SCAN), Hierarchical Dilated Convolutional Block (HDCB), and Residual Path Convolutional Chain (RPCC).**

detection model. The floodgates to the model lie on the been trained on enhanced convolutional learning framework, which stacks on batch normalisation layer after its ReLU activated first layer with a dilation of 1 using 64 3 × 3 convolutional filters; this layer is then followed closely by a MaxPooling layer with a pool size of 2 × 2. Most importantly, the model's performance does not significantly deteriorate by an increase in the number of parameters as local and global diagonal convolutions seem to be covered in the ECLF. Following that, the SCAN module serves its purpose quite efficiently; this module has two attention mechanisms: spatial attention and channel attention. The former requires the kernel to be set at 7 × 7 and only applies a 2D attention map with a single channel to the sigmoid output. In contrast, the latter first performs pooling, which is then set through two biasing-connected layers with 64 set units, and the second layer matches the units to the channels.

For the HDCB, 3 × 3 convolution at 64 filters at a dilated rate of 1, 2, and 3 is performed on the model. This method allows the network to process information from dilated convolutions by combining features extracted at multiple receptive field sizes, facilitating

the capture of local and global contextual information. This method helps the network to use information obtained from the dilated convolutions in a multifaceted manner, as it helps aggregate features obtained from several receptive field sizes. Once the results of all dilated convolution processes have been concatenated, a $1 \times 1$ Convolution having 128 filters is applied on the output with batch normalisation followed by ReLU activation functions to enhance diversity in the production by compressing facets from multiple scales together. The RPCC block supports residual connections in which each block always has two layers of $3 \times 3$ convolutional blocks with 128 kernel filters and a layer of batch normalisation. The output of the last convolutional layer is fed back to the input layer *via* a skip connection, which lets the model preserve such information in deeper layers. Thus, in conjunction with this, the vanishing gradient problem is also solved, and properties are correctly disseminated inside the network.

In the last stage, the model is global average pooled, and then fully dense output layers are incorporated. Standard activation functions and normalization layers were incorporated throughout the architecture to ensure stable and efficient training. There are dense layers of 512 units and 256 units, and both are followed by ReLU activation functions accompanied by layers of dropout, which is set to 0.5 for regularization purposes. Whereas the final layer utilises a single neuron with binary functions and sigmoid activation as it encapsulates jumping off whether sepsis exists or not. This architecture can generalise well across different patients' data while still being able to learn intricate patterns within clinical data at the same time.

### Enhanced convolutional learning framework

The ECLF is the first module in the model architecture, designed to capture both local and global features from medical data using advanced convolutional techniques. It employs atrous convolutions (or dilated convolutions) and a multi-level learning process to learn hierarchical feature representations from the input data. This framework enables the model to capture rich and diverse patterns at different scales, which is critical for detecting complex clinical data patterns that indicate sepsis. Atrous convolution helps in increasing the receptive field without introducing additional parameters. This is especially useful when working with medical time-series data, as it allows the model to capture context from more significant regions without the need for huge kernels. The dilated convolution operation for a feature map $X$ can be defined as:

$$Y_{i,j,k} = \sum_{m,n} X_{i+m\cdot d, j+n\cdot d, k} \cdot W_{m,n,k} \tag{8}$$

where, $Y_{i,j,k}$ is the output of the convolution for the location $(i, j)$ in the feature map, $X_{i,j,k}$ is the input feature map, $W_{m,n,k}$ is the convolution kernel, $d$ is the dilation rate (controls the spacing between kernel elements), $m$ *and* $n$ are the filter coordinates.

The dilation rate $d$ expands the kernel by inserting zeros between its elements, allowing the model to cover a larger area without increasing the number of parameters. This is important for capturing long-range dependencies within the data.

*Multi-level learning*

The ECLF employs a multi-level learning process to capture features at different scales and increase the depth of the feature extraction process. This process applies convolutions with varying kernel sizes to extract information at various levels of granularity. Each framework level is designed to focus on different spatial scales, enabling the algorithm to detect both broad and specific trends in the provided data.

For example, the input feature map $X \in \mathbb{R}^{H \times W \times C}$ is passed through multiple convolutional layers with different kernel sizes, such as:

$$Y_1 = Conv_1(X, K_1) \tag{9}$$
$$Y_2 = Conv_2(X, K_2) \tag{10}$$
$$Y_3 = Conv_3(X, K_3) \tag{11}$$

where, $Y_1, Y_2, Y_3$ are the feature maps obtained after applying convolutions with diverse kernel sizes $K_1, K_2, K_3$, $Conv_i(X, K_i)$ represents the convolution operation applied to the input feature map $X$ with kernel $K_i$, $H,\ W,\ and\ C$ represent the height, width, and number of channels of the input feature map $X$. The multi-level learning process helps the model learn more abstract and complex features, improving its ability to identify sepsis-associated patterns. The features are combined after obtaining feature maps from multiple convolutional layers with different kernel sizes. This can be done through concatenation, aggregating information across the other scales. The resulting feature map is a richer representation of the input data.

$$X_{ECLF} = Concatenate(Y_1, Y_2, Y_3) \tag{12}$$

where $X_{ECLF}$ is the combined feature map after applying the multi-level convolutions. This combination allows the model to maintain fine-grained and high-level features, facilitating a more comprehensive understanding of the data. Non-linear activation functions (ReLU) were applied after convolutional layers to support complex feature transformations. The network is made non-linear by applying non-linear activation functions, like ReLU, after convolutional layers have been applied. This allows the model to learn complex mappings from input to output.

$$X_{ECLF} = ReLU(X_{ECLF}) \tag{13}$$

where $ReLU(x) = max(0, x)$ is the element-wise ReLU function, which sets all negative values to zero and passes positive values unchanged. To stabilise the training process and reduce overfitting, batch normalisation is applied after each convolutional operation. Batch normalisation normalises the output of each layer, ensuring that the network trains faster and more efficiently. The batch normalisation operation is defined as:

$$\hat{X} = \frac{X - \mu_B}{\sigma_B + \varepsilon} \tag{14}$$

where, $X$ is the input to the batch normalisation layer, $\mu_B$ and $\sigma_B$ are the mean and standard deviation of the batch, $\varepsilon$ is a small constant added to prevent division

by zero. The normalised output is then scaled and shifted using learnable parameters $\gamma$ and $\beta$:

$$BN(X) = \gamma \hat{X} + \beta \tag{15}$$

The ECLF effectively learns multi-scale features from the data, enabling the model to detect complex clinical patterns indicative of sepsis. Using dilated convolutions, multi-level learning, and advanced activation functions ensures the model is powerful and efficient for sepsis detection tasks.

### Spatio-channel attention network

The SCAN focuses on the feature learning enhancing process through the regions and channels in the feature map that are most significant. SCAN aspires to refine the attention maps generated by the previous layers of the network by selectively emphasising regions of importance (spatial attention) and channels of importance (channel attention). This allows the model to detect more relevant patterns in the medical data for sepsis identification, thus increasing its discriminative ability. The SCAN comprises two attention schemes: spatial attention mechanism (SAM) and channel attention mechanism (CAM). Approaching it now sequentially, both mechanisms are employed to refine the generated feature map further, thereby increasing the model's ability to discriminate and, vice versa, the unnecessary details.

### Spatial attention mechanism

The SAM aims to assist the model in concentrating on the essential parts of the feature's map. It enhances areas that have relevant information and suppresses the rest. This is crucial in medical data, where some input bits have more information than others. Let $X \in \mathbb{R}^{H \times W \times C}$ be the input feature map, where $H$ is the height, $W$ is the width, and $C$ is the number of channels. The goal of SAM is to generate a spatial attention map $A_{\text{spatial}} \in \mathbb{R}^{H \times W \times C}$ that can be used to weight the feature map spatially.

First, a convolutional layer is applied to the input feature map $X$ to compute the spatial attention map. This operation reduces the channel dimension of the feature map by applying a convolutional filter and then uses a sigmoid activation to obtain the attention map:

$$A_{\text{spatial}}(i,j) = \sigma(Conv_1(Conv_2(X))) \tag{16}$$

where, $A_{\text{spatial}}(i,j)$ is the spatial attention value for each spatial location $(i,j)$, $Conv_1$ and $Conv_2$ are convolutional layers with kernel sizes chosen to capture spatial features, $\sigma$ is the sigmoid activation function applied to ensure the attention map values are between 0 and 1. Once the spatial attention map $A_{\text{spatial}}$ is obtained, the feature map $X$ is multiplied element-wise by $A_{\text{spatial}}$, focusing the model's attention on the most relevant spatial regions:

$$X_{spatial} = X \cdot A_{spatial} \tag{17}$$

*Channel attention mechanism*

The CAM focuses on emphasising the most informative channels of the feature map. Not all channels in the feature map are equally important; some channels carry more valuable information about the presence of sepsis. The goal of CAM is to generate a channel-wise attention map $A_{channel} \in \mathbb{R}^C$, which can be used to reweight the channels of the feature map. The input feature map $X \in \mathbb{R}^{H \times W \times C}$ is first passed through a global average pooling (GAP) operation along the spatial dimensions *H and W* to produce a global summary for each channel:

$$\hat{X}_k = \frac{1}{H \times W} \sum_{i=1}^{H} \sum_{j=1}^{W} X_{i,j,k} \tag{18}$$

where $\hat{X}_k \in \mathbb{R}^C$ is the channel-wise pooled representation, capturing global features for each channel $k$. The pooled feature vector $\hat{X}$ is then passed through a two-layer fully connected (FC) network to compute the channel attention map $A_{channel}$:

$$A_{channel} = \sigma(FC_2(ReLU(FC_1(\hat{X})))) \tag{19}$$

where, $\hat{X}$ is the pooled feature vector, $C_1$ and $C_2$ are fully connected layers, *ReLU* is the activation function applied after the first fully connected layer, $\sigma$ is the sigmoid activation applied to obtain the final channel attention map $A_{channel} \in \mathbb{R}^C$, which contains values between 0 and 1 for each channel. The feature map $X_{spatial}$ from the spatial attention mechanism is then multiplied element-wise by the channel attention map $A_{channel}$, focusing the model's attention on the most informative channels:

$$X_{final} = X_{spatial} \cdot A_{channel} \tag{20}$$

The SCAN module applies spatial and channel attention mechanisms to refine the input feature map in a two-stage process. By zeroing in on the most important areas, the spatial attention process improves the feature map. By highlighting the most important channels, the channel attention mechanism continuously adjusts the feature map. The final refined feature map $X_{final}$ is the result of applying both attention mechanisms sequentially:

$$X_{final} = (X \cdot A_{spatial}) \cdot A_{channel} \tag{21}$$

This refined feature map is then passed on to the next module in the network, which may be a dilated convolutional block or residual path convolutional chain, to process the features further for sepsis detection.

### Hierarchical dilated convolutional block

The HDCB is a powerful tool for obtaining hierarchical features crucial for recognising low-level patterns associated with sepsis in medical data sets. HDCB considers local and global contexts concurrently by utilising stacked dilated convolutions, thereby processing input at different scales. The combination of parameters for the convolutional neural networks with dilation rates enables the model to learn multi-level complexities affordably. From a mathematical perspective, the model can cover a wider overall field with minimal

computational resources. In summation, HDCB utilises convolutional layers possessing multi-dilation rates to achieve a state-of-the-art contraction. A standard convolution operation with a kernel of size $K \times K$ applied to an input feature map $X \in \mathbb{R}^{H \times W \times C}$ can be expressed as:

$$Y_{i,j,k} = \sum_{m,n} X_{i+m,j+n,k} \cdot W_{m,n,k} \tag{22}$$

where, $Y_{i,j,k}$ is the output feature map, $W_{m,n,k}$ is the kernel weight, $(m, n)$ are the coordinates of the kernel. For dilated convolutions, the dilation rate $d$ introduces spacing between the kernel elements. The dilated convolution operation is defined as:

$$Y_{i,j,k} = \sum_{m,n} X_{i+m.d,j+n.d,k} \cdot W_{m,n,k} \tag{23}$$

where, $d$ is the dilation rate that controls the spacing between kernel elements. The dilation rate allows the model to capture larger contexts without increasing the size of the kernel. For example, a dilation rate of $d = 2$ will effectively double the receptive field of the kernel, capturing more distant dependencies within the feature map. The HDCB stacks several dilated convolutional layers to capture information at multiple scales, each with a different dilation rate. Let's consider $N$ dilated convolutional layers, each with a different dilation rate $d_1, d_2, \ldots, d_N$. The output of each layer can be written as:

$$Y_1 = Conv_1(X, d_1) \tag{24}$$
$$Y_2 = Conv_2(X, d_2) \tag{25}$$
$$Y_N = Conv_N(X, d_N) \tag{26}$$

where, $Y_1, Y, \ldots, Y_N$ are the outputs of each dilated convolutional layer, $Conv_1(X, d_1)$ represents the convolution operation with a dilation rate $d_i$ applied to the input $X$. Each convolutional layer captures different levels of context, with more significant dilation rates focusing on more global information and smaller dilation rates capturing local details. The outputs from the stacked dilated convolutional layers are then concatenated or summed to combine the features from different scales. The aggregated feature map $X_{HDCB}$ can be obtained as:

$$X_{HDCB} = Concatenate(Y_1, Y_2, \ldots, Y_N). \tag{27}$$

Alternatively, a summation approach can be used:

$$X_{HDCB} = Y_1 + Y_2 + \cdots + Y_N. \tag{28}$$

This step combines the features learned at different scales, resulting in a richer and more hierarchical representation of the input data. The model can learn more abstract features essential for detecting sepsis by capturing local and global information. After the dilated convolutions and feature aggregation, non-linear activation functions and Batch normalization are applied. The activation function is applied element-wise:

$$X_{HDCB} = ReLU(X_{HDCB}) \tag{29}$$

where $ReLU(x) = max(0, x)$. Batch normalisation is applied to the aggregated feature map to normalise the activations, improving the convergence speed and preventing overfitting. The batch normalisation operation is defined as:

$$\hat{X} = \frac{X - \mu_B}{\sigma_B + \varepsilon} \tag{30}$$

where, $\mu_B$ and $\sigma_B$ are the mean and standard deviation of the batch, $\varepsilon$ is a small constant added to prevent division by zero. The normalised output is then scaled and shifted using learnable parameters $\gamma$ and $\beta$:

$$BN(X) = \gamma\hat{X} + \beta. \tag{31}$$

After applying the dilated convolutions, feature aggregation, activation, and normalisation, the final output of the HDCB is the refined feature map $X_{HDCB}$. This feature map contains hierarchical information from local and global contexts, enabling the model to learn complex patterns indicative of sepsis.

$$X_{HDCB} = ReLU\left(BN\left(\sum_{i=1}^{N} Y_i\right)\right). \tag{32}$$

The HDCB enhances the model's ability to capture long-range dependencies and subtle patterns in medical data, making it particularly effective for detecting sepsis in clinical environments.

### Residual path convolutional chain

The RPCC is one of the model components meant to facilitate the information flow through the network and make it easier to train intense networks. It uses their residual connections, also called skip connections, to address the vanishing gradient problem and to aid the learning of complex representations. The RPCC architecture reduces feature overfitting by permitting more efficient gradient flow to the feature maps during backpropagation, which allows for effective learning by the model. In RPCC, the feature map undergoes several convolutional operations, and a skip connection is made after each operation. These connections help a convolutional layer learn the residual over that layer alongside identity mapping while several layers are bypassed, making it more flexible to fit intricate structures. A residual connection allows input to cross single or multiple layers and then be summed up in the output, providing a significantly direct path. This contributes to retaining important data over layers, such as in deep networks. For a feature map $X$, the residual connection is defined as:

$$X_{residual} = X + F(X) \tag{33}$$

where, $X$ is the input feature map, $F(X)$ represents the output of the convolutional operation applied to $X$. The function $F(X)$ is typically a sequence of convolutional layers, batch normalisation, and activation functions, and can be expressed as:

$$F(X) = Conv(ReLU(BN(Conv(X)))) \tag{34}$$

where, *Conv* represents a convolutional layer, *BN* denotes batch normalisation, *ReLU* is the activation function. Thus, the residual connection allows the network to learn an additional mapping $F(X)$, while also maintaining the original input $X$ directly. The RPCC is built by stacking multiple convolutional layers, each with its residual connection. Given an input feature map $X$, the output of the first residual block is:

$$X_1 = X + F_1(X) \tag{35}$$

where $F_1(X)$ is the output of the first convolutional block, which includes convolution, batch normalisation, and activation. The input is passed through the residual connection for subsequent residual blocks and updated at each step. The output of the second residual block is:

$$X_2 = X_1 + F_2(X_1) \tag{36}$$

where $F_2(X_1)$ is the output of the second convolutional block, and so on. This can be generalized for $N$ residual blocks as:

$$X_N = X_{N-1} + F_N(X_{N-1}) \tag{37}$$

where $F_N(X_{N-1})$ is the output of the $N$-th convolutional block. The final output feature map is produced after passing through all the residual blocks. The RPCC ensures that essential features are preserved and propagated throughout the network. The final output of the RPCC block, denoted as $X_{RPCC}$, is:

$$X_{RPCC} = X_N \tag{38}$$

where $X_N$ is the final feature map after applying all residual connections and convolutional blocks. An activation function and batch normalisation follow each convolutional operation in the residual path to introduce non-linearity and stabilise the learning process. The output of each convolutional layer is first passed through a ReLU activation to introduce non-linearity:

$$X_{activated} = ReLU(X_{RPCC}). \tag{39}$$

Batch normalisation is applied to reduce internal covariate shift and ensure more stable training:

$$\hat{X} = \frac{X - \mu_B}{\sigma_B + \varepsilon} \tag{40}$$

where, $\mu_B$ and $\sigma_B$ are the mean and standard deviation of the batch, $\varepsilon$ is a small constant to prevent division by zero. Finally, the normalised output is scaled and shifted using learnable parameters $\gamma$ *and* $\beta$ :

$$BN(X) = \gamma\hat{X} + \beta. \tag{41}$$

In some cases, especially when the dimensions of the input and output feature maps are the same, the residual connection is an identity mapping, meaning the input is directly added to the output without any transformation. This is particularly useful in maintaining

information across layers in very deep networks. In this case, the residual connection is simply:

$$X_{residual} = X. \tag{42}$$

After passing through several residual layers, the final feature map $X_{RPCC}$ can be obtained from the last residual block. The complete residual chain can be written as:

$$X_{RPCC} = X + F_11(X) + F_2(X1) + \cdots + F_N(X_{N-1}). \tag{43}$$

The output $X_{RPCC}$ is the enriched feature map containing preserved and learned features across multiple residual paths. The Residual Path Convolutional Chain (RPCC) ensures efficient feature propagation. It enables the network to learn deeper, more complex features without the risk of vanishing gradients. This makes it particularly useful for detecting subtle patterns in data, such as those associated with sepsis detection.

### African vulture optimization algorithm pseudocode

The AVOA is a nature-inspired optimization algorithm that mimics the foraging behavior of vultures. Vultures search for food in an environment by utilizing a combination of exploration and exploitation strategies.

## RESULT AND DISCUSSION

This section discusses the performance of the proposed model based on the architecture integrating ECLF, SCAN, HDCB, and RPCC. In particular, ResNet was evaluated on a sizeable clinical dataset.

As part of the model development procedure, the dataset was divided into three subsets: the test set, the train set, and the validation set. The model parameters were updated using the defined training set during the learning phase. About the learning phase, the validation set was used to monitor the model's performance and control guided hyperparameter tuning and overfitting using early stopping strategies on the model during each defined epoch. This form of intermediate assessment is termed validation accuracy and is plotted on a per-epoch basis in the respective graphs. However, the test set was exclusively held to evaluate the model after its training. Once the model's training was completed, the model's final configuration was assessed using the test set. This evaluation is known as generalization evaluation and is performed only once at the end of training. The performance obtained at this stage is referred to as testing accuracy, which is calculated to show the model's performance on completely new data.

This model outperformed the rest with an accuracy of 99.4%, which proves the model's efficacy in detecting sepsis at a precocious stage. Such models emerge as competent predictive models since they accommodate and utilise robust multi-scale feature extraction and attention mechanisms to enhance the model's accuracy. Hence, the model trained demonstrates competitive and outstanding predictions.

The loss and accuracy values for the model's training and validation were tracked throughout epochs. The training accuracy kept on enhancing and reached a commendable 99.4% mark during the final phase of the training. Alongside this, validation accuracy continued to depict a similar trend, hence indicating that the model in question is not

**Algorithm 1 African vulture optimization algorithm.**

1: Initialise population $P$ of vultures (each vulture represents a potential solution)
2: Set the maximum number of iterations ($max_{iter}$)
3: Set the maximum number of vultures ($population_{size}$)
4: Set parameters for exploration and exploitation phases:
5:     Step size ($step_{size}$)
6:     Exploration radius ($exploration_{radius}$)
7:     Exploitation factor ($exploitation_{factor}$)
8:     Reproduction rate ($reproduction_{rate}$)
9:     Breeding factor ($breeding_{factor}$)
10: Define $fitness\_function(x)$
11: $best_{solution} \leftarrow \infty$
12: $best_{fitness} \leftarrow \infty$                                    ▷ Main AVOA loop
13: **for** $it\ er$ in $range(max_{iter})$ **do**          ▷ Evaluate the fitness of each vulture in the population
14:     **for** $i$ in $range(population_{size})$ **do**
15:         $vulture \leftarrow P[i]$
16:         $fitness \leftarrow fitness_{function}(vulture)$        ▷ Update the best solution found so far
17:         **if** $fitness < best_{fitness}$ **then**
18:             $best_{solution} \leftarrow vulture$
19:             $best_{fitness} \leftarrow fitness$
20:         **end if**
21:     **end for**          ▷ Apply foraging and scavenging behaviour (exploration phase)
22:     **for** $i$ in $range(population_{size})$ **do**          ▷ Random exploration: move vulture randomly
23:         **if** $random() < exploration_{radius}$ **then**
24:             $P[i] \leftarrow P[i] + random_{step}(step_{size})$
25:         **else**                                    ▷ Scavenging: exploit promising areas
26:             $P[i] \leftarrow P[i] + exploitation_{factor} * (best_{solution} - P[i])$
27:         **end if**
28:     **end for**                                    ▷ Apply reproduction and breeding behaviour (exploitation phase)
29:     **for** $i$ in $range(best_{solution})$ **do**
30:         **if** $random() < reproduction_{rate}$ **then**          ▷ Breeding: combine features of two solutions to create a new solution
31:             $parent1 \leftarrow P[i]$
32:             $parent2 \leftarrow P[random_{choice}(population_{size})]$
33:             $P[i] \leftarrow breeding_{factor} * (parent1 + parent2)/2$
34:         **end if**
35:     **end for**          ▷ Optionally apply a local search or mutation strategy to refine solutions further

(Continued)

| Algorithm 1 (continued) |
|---|
| 36:   **for** $i$ in $range(population_{size})$ **do** |
| 37:       $P[i] \leftarrow local_{search}(P[i])$ |
| 38:   **end for** |
| 39: **end for** |
| 40: **return** $best_{solution}$ |

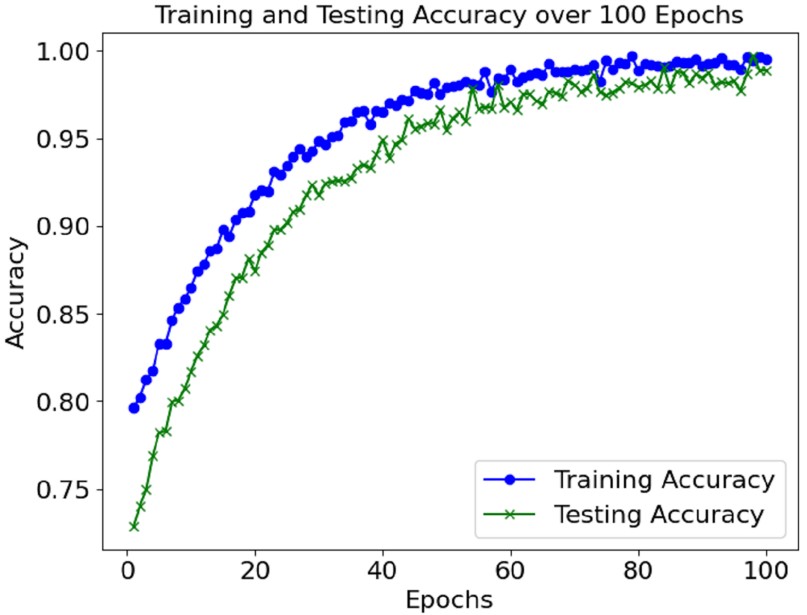

**Figure 2  Accuracy graph of the proposed model.** 

overfitting. Until a certain point, the training and validation loss started low and subsequently grew to a point where it reached a loss minimum, which predicts the model's convergence.

Figure 2 shows the accuracy of the training and validation sets over the 100 epochs. The model performed well and maintained a high level of accuracy, with very little change in the validation accuracy during the training.

The training and validation losses also decreased, as depicted in Fig. 3. The convergence of the loss values is consistent enough to demonstrate that the model has learned sufficiently and is not overfitting the training set. Additional measures, including precision, recall, F1-score, and AUC, were used to evaluate the model's performance and reported accuracy. Such metrics help determine the model's effectiveness in distinguishing between positive (sepsis) and negative (non-sepsis) conditions, which is essential in medical settings and is depicted in Table 1.

The performance evaluation from the proposed model is exhaustively outlined in Table 1. Accuracy achieved by the model is exceptionally high at 99.4%, displaying effectiveness for precise classification of sepsis and non-sepsis cases. The model's precision

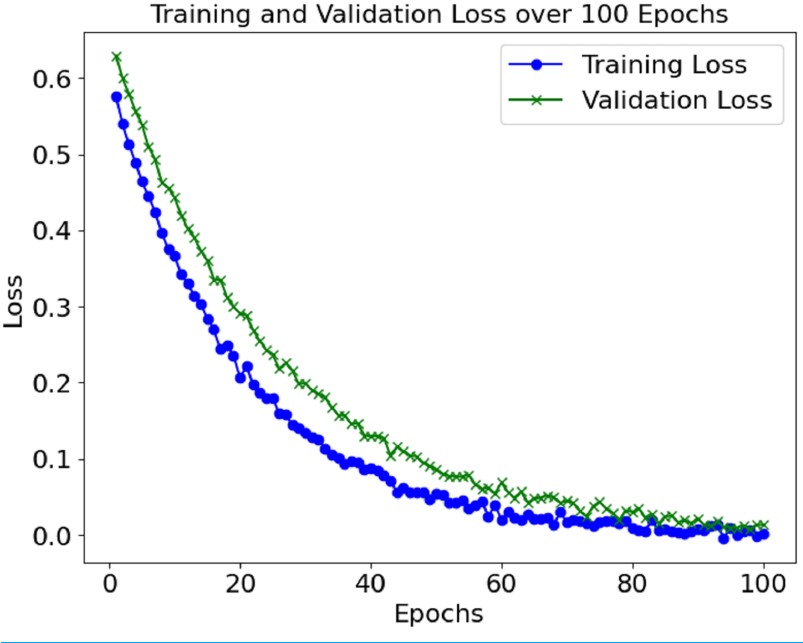

**Figure 3 Loss graph of the proposed model.**

**Table 1 Performance analysis.**

| Metric | Value |
| --- | --- |
| Accuracy | 99.4% |
| Precision | 98.7% |
| Recall | 99.2% |
| F1-Score | 99.0% |
| AUC | 0.998 |
| Specificity | 99.3 |
| False Positive Rate (FPR) | 0.7 |
| False Negative Rate (FNR) | 0.8 |

of 98.7% indicates that the model exercises strong control over false positives where patients diagnosed with sepsis are minimized, and is vital to prevent unnecessary interventions. The model demonstrated a recall of 99.2% which shows high sensitivity in identifying the actual sepsis cases, proving vital for diagnosis and treatment. The balanced F1-score of 99% illustrates that the proposed model also maintained a well-balanced trade-off between precision and recall, reaffirming the strong figure reported for recall. AUC value of 0.998 provides further proof the model has strong discriminatory power in distinguishing between sepsis and non-sepsis conditions. Furthermore, the model achieved 99.3% specificity in correctly classifying non-sepsis patients while maintaining an extraordinary low false positive rate of 0.7% and false negative rate of 0.8%.

In Fig. 4, this section presents a confusion matrix associated with the performance outcomes of the model offered in this article for the detection of sepsis. The matrix

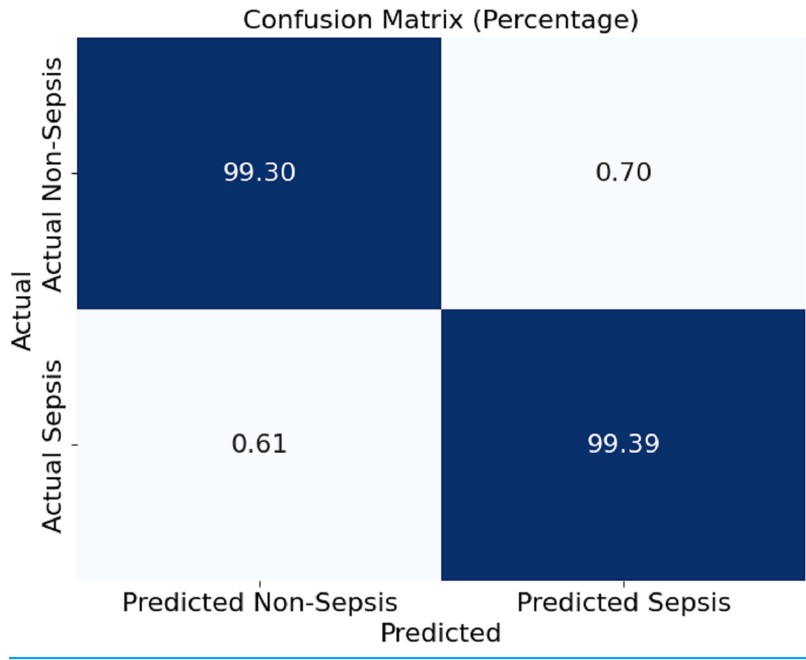

**Figure 4 Confusion matrix of the proposed model.**

contains values of correct predictions as considerations of matrices are normalised to percentages. The model showed 99.30% and 99.39% of efficiency in distinguishing the cases of non-sepsis and the cases of sepsis, respectively. It can be noted that false negative rates of 0.61% and false positive rates of 0.70%, as the off-diagonal numbers, mean that the classification has a mild depth of error for the model, which is even further corroborated by the model's ability to detect sepsis and non-sepsis cases accurately.

The relationship between accuracy and hyperparameters are represented in Fig. 5 as well. The learning rate on the left orthogonal graph shows that the overall accuracy goes hand in hand with the rate until reaching somewhere about 0.01, at this point, performance wisely shifts down and can be observed as the learning goes further. The right graph Instead demonstrates that accuracy loses its power through increments of dropout from 0.5. On the other hand, the graph suggests a way out, controlling the moderato level of dropout, as it proved to augment model generalisations. However, a performance a wee higher than 0.5 could cause impairing effects. Finally, these results show and exemplify how, in the realm of model performance tuning, both the learning and dropout rates can prove to be crucial.

The model outlined in the previous sections achieved a perfect score of AUC = 1.00, as shown in Fig. 6. Actual positive rate (TPR) and the true negative rate (TPR) were plotted on the y-axis and x-axis, respectively. The ROC region is very far from the dotted diagonal. Therefore, the model is good at differentiating between different diseases and non-use cases, with sepsis being one of them. Achieving perfect scores against AUC as high as one suggests that the evaluated model has achieved high accuracy in differentiating between positive and negative cases. Through a methodical process of module removal or

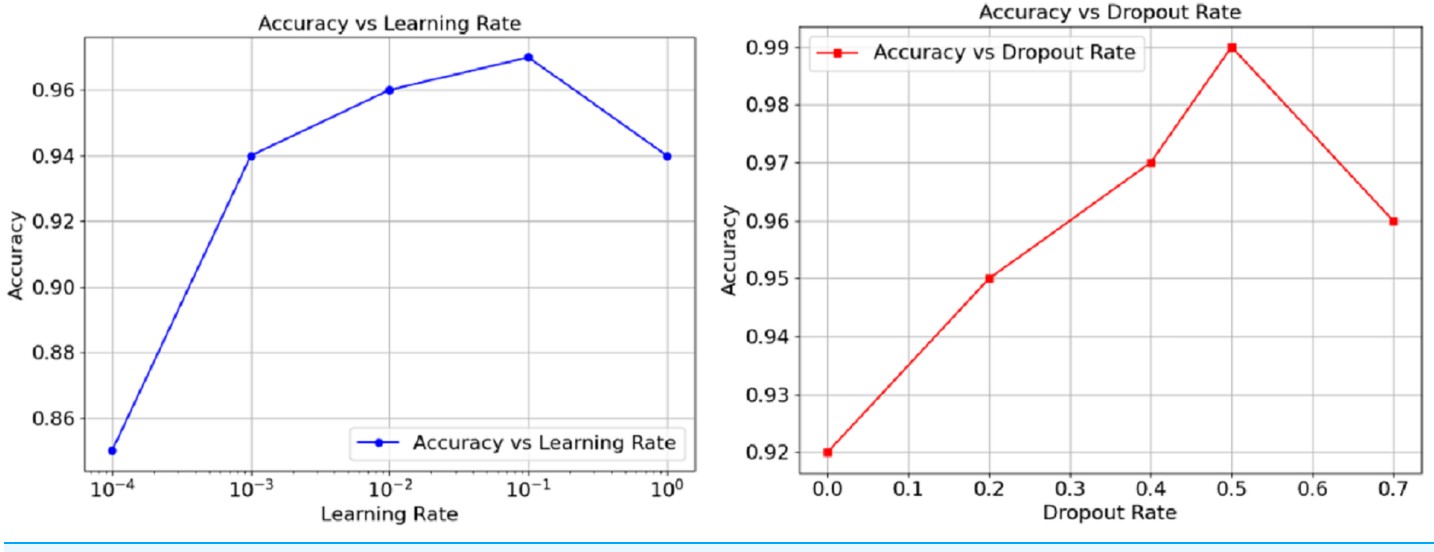

**Figure 5 Accuracy *vs*. Dropout rate and learning rate.**

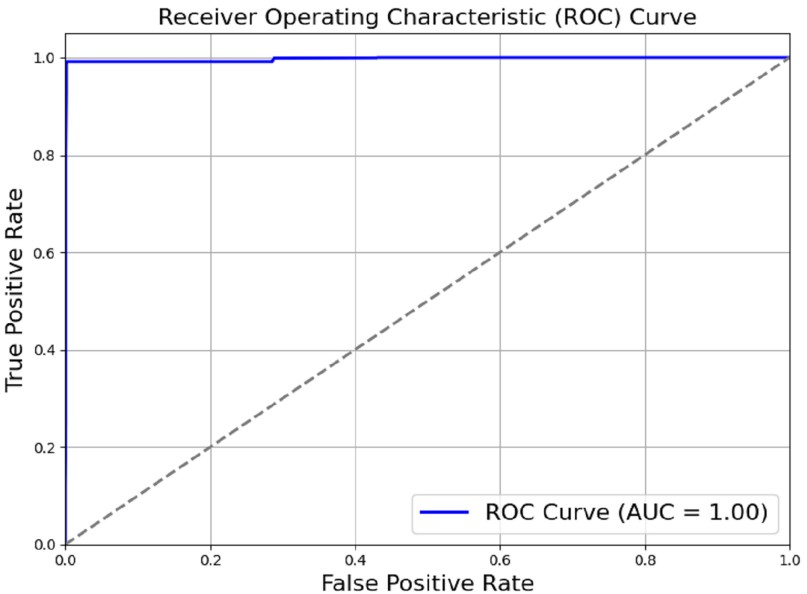

**Figure 6 ROC_AUC of the proposed model.**

configuration modification, we conducted ablation research to evaluate the contribution of each model component. Table 2 summarises the ablation study's findings by comparing the full model to subsets of the model that had their components reduced.

To ensure that the model was robust, we ran a 6-fold cross-validation on the dataset. To train and test the model, we divided the dataset into six sections, and we did it six times, every single time using an alternate fold as the validation set and the other five as the training set. The results of cross-validation are shown in Table 3.

**Table 2 Ablation study results.**

| Model variant | Accuracy | Precision | Recall | F1-score | AUC |
|---|---|---|---|---|---|
| Full model | 99.4% | 98.7% | 99.2% | 99.0% | 0.998 |
| Without SCAN | 97.8% | 96.2% | 97.4% | 96.8% | 0.985 |
| Without HDCB | 98.5% | 97.5% | 98.0% | 97.7% | 0.993 |
| Without RPCC | 97.1% | 95.8% | 96.5% | 96.1% | 0.986 |
| Without ECLF | 96.3% | 94.6% | 95.5% | 95.0% | 0.978 |

**Table 3 6-Fold cross-validation results.**

| Metric | Mean (%) | Standard deviation (%) |
|---|---|---|
| Accuracy | 99.38% | ±0.14 |
| Precision | 98.78% | ±0.32 |
| Recall | 99.22% | ±0.27 |
| F1-score | 99.0% | ±0.25 |
| AUC | 0.9978% | ±0.0011 |

The results of six-fold cross-validation are summarized in Table 3. The proposed model achieved a mean accuracy of 99.38% with a standard deviation of ±0.14%, indicating high consistency across different data splits. Similarly, the precision, recall, and F1-score maintained low variability, further demonstrating the robustness of the model. The AUC remained exceptionally high at $0.9978 \pm 0.0011$, highlighting the model's strong discriminative ability for early sepsis detection. To test the performance of the model designed during this study and compare its performance against other popularly known techniques for detecting sepsis, a comparison was carried out of the model with the rest. The models compared include logistic regression (LR), support vector machine (SVM), random forest (RF) models, which are some basic models, and the CNN, which is also fundamental. The results of the comparison are included in Table 4.

To analyze the effectiveness of the proposed hybrid deep learning model, it was compared to existing models through evaluation of other machine learning and deep learning methods. As presented in Table 4, the performance of the proposed model is compared against benchmarks stemming from logistic regression, SVM, random forest, and a baseline CNN model. The proposed model significantly outperformed all existing methods across all critical metrics. The recall and precision metrics of the logistic regression and SVM models were 85.7% and 92.6% respectively; however, these measures are not adequate in high-stakes medical environments where detection must be both timely and accurate. Even strong classifiers such as RF and the baseline CNN model were incapable of surpassing 94.8% and 96.7% accuracy, respectively. Unlike the other algorithms, the proposed model achieved 99.4% accuracy, 98.7% precision, 99.2% recall, and 0.998 AUC, which signifies a remarkable performance benchmark exhibiting dependability in predictive accuracy alongside sensitivity and specificity necessary to lower the rate of erroneous diagnosis in sepsis cases. Considerable increase in performance can

**Table 4 Results of performance comparison.**

| Model | Accuracy | Precision | Recall | F1-score | AUC |
|---|---|---|---|---|---|
| Proposed model | 99.4% | 98.7% | 99.2% | 99.0% | 0.998 |
| Logistic regression | 85.7% | 81.3% | 87.0% | 84.1% | 0.758 |
| SVM | 92.6% | 91.4% | 93.1% | 92.2% | 0.873 |
| Random forest | 94.8% | 93.0% | 95.4% | 94.2% | 0.896 |
| CNN (Simple) | 96.7% | 95.8% | 97.2% | 96.5% | 0.957 |

be linked to the addition of new components like the SCAN, HDCB, RPCC, and the hyperparameter optimization executed with the AVOA. Such integration allows the model to capture sophisticated, non-linear relations in multi-dimensional clinical data which other conventional models would not identify.

In Table 5 we compare the results of our proposed model with multiple reported state-of-the-art models with applied methods of prediction on sepsis. The results clearly demonstrate the accuracy challenges faced by conventional machine learning approaches and even earlier deep learning models from the years 2017 and 2018 which are dominated by the complex, nonlinear, temporal, and intricate nature of sepsis progression. As an example neural hybrid models composed from the convolutional and recurrent networks integrated the use of traditional deep learning models, but their performance remained stagnant, yielding predictive accuracy of 56.25% (*Duan et al., 2023*). Followed by this, *Strickler et al. (2023)* reported a slightly improved accuracy (75.00%) with the use of LSTM networks. Most of these results are lower than baseline, indicating that there is still a long journey ahead before these models can achieve clinically acceptable predictive performance. While results achieved by more advanced models tend to perform better, that's not an argument that ignores the performance forecasts of cutting edge models. In either case, *Zhou, Beyah & Kamaleswaran (2021)* OnAI-Comp, and *Rosnati & Fortuin (2021)* with their attention time convolutional network (AttTCN), managed to attain accuracy scores of 81.25% and 68.75%. These findings highlight the significance of attention mechanisms and temporal modeling for multi scale feature extraction frameworks. On the other hand, the hybrid deep learning model outperforming the rest achieves 99.4% accuracy, a figure far exceeding the findings of prior studies. This is because the model uses the ECLF which deals with spatial feature extraction, the SCAN which attends to noteworthy clinical features on the fly, and the RPCC which retains key information across deep network layers. Also, the model's performance was optimally complemented with hyperparameter adjustments *via* the AVOA. This underscores the novelty of the approach since the model was designed with extensive focus on increasing accuracy while solving the practical clinical problem of reliably and promptly detecting the onset of clinical sepsis.

## Limitations and future work

While the proposed model demonstrates outstanding performance on the Kaggle Sepsis Prediction dataset, this study has certain limitations that must be acknowledged. The

**Table 5 Comparison with the state of the art techniques.**

| Reference | Methodology | Accuracy (%) |
| --- | --- | --- |
| *Gholamzadeh, Abtahi & Safdari (2023)* | Gaussian naïve Bayes (NB), decision tree (DT), random forest (RF) | 68.75 |
| *Duan et al. (2023)* | CNN + RNN | 56.25 |
| *Strickler et al. (2023)* | LSTM | 75.00 |
| *Zhou, Beyah & Kamaleswaran (2021)* | Online Artificial Intelligence Experts Competing Framework (OnAI-Comp) | 81.25 |
| *Al-Mualemi & Lu (2020)* | RNN-LSTM, SVM | 62.50 |
| *Nemati et al. (2018)* | Attention time convolutional network (AttTCN) | 68.75 |
| Proposed methodology | ECLF, SCAN, HDCB, and RPCC | 99.4 |

experimental evaluation is confined to a single publicly available dataset, which may limit the generalizability of the results across diverse clinical environments and patient populations. Variations in data quality, patient demographics, clinical protocols, and sensor accuracy across different healthcare institutions could impact the performance of the model when applied to other datasets. Additionally, although a detailed comparison with several state-of-the-art model's performance models using the same dataset has been provided, further validation against other benchmark datasets such as the MIMIC-III or MIMIC-IV clinical databases would strengthen the evidence for the model's robustness and adaptability. Future research will focus on validating the proposed model on multiple datasets from varied clinical settings to ensure broader applicability. Incorporating real-time streaming data from electronic health records (EHRs) and evaluating the model in prospective clinical trials will also be explored to assess its performance in practical deployment scenarios. Moreover, enhancing model interpretability and explainability through integration with explainable AI (XAI) techniques will be a critical area of future development to support clinical decision-making effectively.

This study enhances AI systems intended for the early detection of critical healthcare conditions, especially sepsis, one of the leading causes of mortality globally in the ICU. Although the model suggests better accuracy than other models, it is worth noting that an early diagnosis does not resolve the multifaceted issues concerning the management of sepsis. Incorporating predictive models into clinical workspaces, real-time data provisioning, and actionable clinical decision support still pose challenges. In addition to the lack of diverse datasets, the explainability and interpretability of the model's predictions should be prioritized in future efforts. Adopted XAI frameworks such as SHapley Additive exPlanations (SHAP) and Local Interpretable Model-agnostic Explanations (LIME) could enable medical professionals to appreciate the determinants of the model's reasoning, thus strengthening their confidence in an AI-assisted system. Also, model performance, especially in the early detection of sepsis, can be improved by addressing data imbalance using advanced focal loss techniques or synthetic data generation like SMOTE.

Another critical aspect is the real-time model validation within clinical settings through live EHR data streams. This will assess the model's latency and efficiency and its impact on

clinically relevant outcomes. Studying the incorporation of multi-modal data sets such as genomic, proteomic, and imaging data may enhance the understanding of sepsis pathophysiology and enable the development of earlier and more precise diagnostic models. Lastly, investigations should be conducted on the ethical issues and potential biases in predictive models designed for other healthcare functions. Addressing equity, clarity, and regulatory alignment concerns will be essential for integrating and functioning AI-powered solutions in everyday medical practice.

## CONCLUSION

The article presents a new method for sepsis detection that is deep learning-based, achieved by ECLF, SCAN, HDCB, and RPCC. These core components act as a hybrid model proven to perform well, achieving an accuracy of 99.4% alongside precision, recall, and F1-score. Due to sepsis detection accuracy, this model dethrones the previous traditional clinical scoring systems alongside baseline ML systems. The ablation study had different components due to model parameter tuning and revealing factors. Combining the ROC curve and the confusion matrix effectively determined the correct classification with the AUC score and its negligible misclassifications. This study proves that improved decision-making alongside early detection modes for sepsis management can lead to a better patient outcome and a lower mortality rate.

Future research could examine incorporating patient demographic and medical history data, wearable sensors, and real-time monitoring into the model to enhance its forecasting ability. However, while the model is robust and had a positive outcome in the controlled experiment, it would need to be validated across several centres to determine if it is widely applicable to numerous patients and supported in a multi-facility setting. Despite achieving exceptional predictive performance, the study's conclusions are drawn based on results from a single dataset. While this provides a strong foundation, further evaluations across diverse datasets and real-world environments are essential to confirm the model's generalizability. Addressing these limitations in future work will ensure that the proposed system can be reliably adopted in varied clinical contexts, ultimately contributing to improved patient outcomes through early and accurate sepsis detection.

### Funding

This work was supported by the Deanship of Research and Graduate Studies at King Khalid University through Large Research Project under grant number RGP2/219/46. Princess Nourah bint Abdulrahman University Researchers Supporting Project number (PNURSP2025R809), Princess Nourah bint Abdulrahman University, Riyadh, Saudi Arabia. This work was also supported by the Deanship of Scientific Research at Northern Border University, Arar, KSA, through the project number "NBU-FFR-2025-2248-05" and the Deanship of Scientific Research at Majmaah University under Project No. R-2025-

1825. This study is supported *via* funding from Prince Sattam bin Abdulaziz University project number (PSAU/2025/R/1446). The funders had no role in study design, data collection and analysis, decision to publish, or preparation of the manuscript.

## Grant Disclosures
The following grant information was disclosed by the authors:
Deanship of Research and Graduate Studies at King Khalid University: RGP2/219/46.
Princess Nourah bint Abdulrahman University, Riyadh, Saudi Arabia: PNURSP2025R809.
Deanship of Scientific Research at Northern Border University, Arar, KSA: NBU-FFR-2025-2248-05.
Deanship of Scientific Research at Majmaah University: R-2025-1825.
Prince Sattam bin Abdulaziz University: PSAU/2025/R/1446.

## Competing Interests
The authors declare that they have no competing interests.

## Author Contributions
- Ahmed S. Almasoud conceived and designed the experiments, analyzed the data, authored or reviewed drafts of the article, and approved the final draft.
- Ghada Moh Samir Elhessewi conceived and designed the experiments, analyzed the data, authored or reviewed drafts of the article, and approved the final draft.
- Munya A. Arasi conceived and designed the experiments, performed the experiments, analyzed the data, prepared figures and/or tables, authored or reviewed drafts of the article, and approved the final draft.
- Abdulsamad Ebrahim Yahya conceived and designed the experiments, analyzed the data, authored or reviewed drafts of the article, and approved the final draft.
- Menwa Alshammeri performed the experiments, analyzed the data, performed the computation work, prepared figures and/or tables, and approved the final draft.
- Donia Badawood performed the experiments, performed the computation work, prepared figures and/or tables, and approved the final draft.
- Faisal Mohammed Nafie performed the experiments, performed the computation work, prepared figures and/or tables, and approved the final draft.
- Mohammed Assiri performed the experiments, performed the computation work, prepared figures and/or tables, and approved the final draft.

## Data Availability
The Prediction of Sepsis dataset is available at Kaggle:
https://www.kaggle.com/datasets/salikhussaini49/prediction-of-sepsis. The dataset and code are available in the Supplemental Files.

## Supplemental Information
Supplemental information for this article can be found online at http://dx.doi.org/10.7717/peerj-cs.2958#supplemental-information.

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
