# Peer review of "Efficient sepsis detection using deep learning and residual convolutional networks"

_PeerJ Computer Science, doi:10.7717/peerj-cs.2958_

## Round 0.1 · original submission · Major Revisions

The article has some value but cannot be accepted in its current form. The reviewers raised a number of issues that need to be addressed by the authors. Please consider these issues and prepare a new enhanced version of the manuscript.

**Language Note:** The review process has identified that the English language must be improved. PeerJ can provide language editing services - please contact us at [email protected] for pricing (be sure to provide your manuscript number and title). Alternatively, you should make your own arrangements to improve the language quality and provide details in your response letter. – PeerJ Staff

Reviewer 1 ·

Basic reporting

It's a well written introductory section. However, in the later section, I cannot see a similar trend.

Experimental design

The scope and importance of research is not adequately clarified. The simple reader does not understand why this subject is important and what is the significance of new research results on such real-world applications, as they appear in our everyday lives.

Validity of the findings

There is no contrast of the latest approach introduced to other current methods in the results section (unless I have ignored).

Additional comments

I will recommend a major revision to reorganize the revised paper:
1. It's a well written introductory section. However, in the later section, I cannot see a similar trend.
2. There are some clear constraints in the introductory section on whether a mixed-method strategy can be used. I would also recommend that the dissertation be focused on current shortcomings and differences.
3. The scope and importance of research is not adequately clarified. The simple reader does not understand why this subject is important and what is the significance of new research results on such real-world applications, as they appear in our everyday lives.
4. There is no contrast of the latest approach introduced to other current methods in the results section (unless I have ignored).

·

Basic reporting

Thank you for providing an important study to sepsis detection. Could you make it clear where/when the detection comes in place as you motivate in the beginning with "if not treated in time". So, can you clarify how your study helps to improve detection (in a timely manner) or does it focus on robustness? It is well said that standard approaches have shortcomings (in terms of empirical findings), however, your manuscript would benefit from a more direct motivation.

Although the language is fairly readable and in well-written English, I find it can have a rework in terms of general language and grammar. There are a lot of sentences that span over 3 lines and claim more than one statement, it would benefit from splitting sentences into two concise and shorter sentences.

Especially your first sentences (introduction, abstract) are this kind of nested.

Mentioning only one particular metric/measurement in the abstract while not mentioning all the others looks weird, so either provide all results or be abstract.

Typos: Développment (!), lower case title at 3.3),
Spacing and alignment: please tidy up your lines 328-330 regarding commas, spaces and topics. 454, 457 "Where," as well.

Experimental design

You mix methods and results, there are measurements and result values listed in the method section. Please stick to describing and not evaluating right away.

Please provide clearer information in dataset details what dataset it is. Only telling "kaggle dataset" is highly unspecific.

Formula (1) is wrongly stated, it uses index i on both sides, i for the missing value and i for the sum index. As well, xi should be excluded from calculating mean values if xi is not present.

Lines 250 to 253 are unnecessary, this is not a study showing/proofing one-hot encoding which is quite a standard in machine learning. Please remove or re-write as it is also confusing the reader using this example (twice). In my opinion it is enough to state you used one-hot encoding for categorial/nominal values, Z-Score for your others. Remember, you are not conducting a lecture in this manuscript.

Although I appreciate you're working out in a detailed manner each step of your model/network, please focus on describing your achievments/modifications. You are tending to overexposes state of the art technology. For example, in 2025 there is no need to explain "using ReLU or any other activation function" as novelty in order to achieve complex input to output transformations. Again, this is state of the art since 10+ years, your manuscript does not benefit from "teaching" about standards like activation functions. If you were using more specific activation function (not "just" std. ReLU) that would impact your outcomes, please let us know.

More important what is missing throughout the results and experimental design is:
how were your testings conducted, how many datasets did you use for training, how many did you use for testing / validation of your findings and to assess accuracy. Table 1: performance analysis is too unspecific just listing accuracy values, please provide more insight.

In the manuscript you refer to validation accuracy, in your figures you name testing accuracy. As you have throughout the training process individual validation steps that you also use to determine the next training epoch. That means, testing-results cannot be tracked throughout the process, they are only applied once in order to assess the (final) model's performance. So, please make this process clearer and what terms you used in which context.

Validity of the findings

Cannot be assessed as there is a lot of details on the experimental setup missing. Without that I cannot assess validity of the findings.

Please provide the information mentioned before, like dataset sizes, how much data you withheld, how your stratification process has been to ensure un-biased testing data and so on.

Reviewer 3 ·

Basic reporting

The manuscript employs clear and unambiguous professional English throughout, which is commendable. However, while the introduction and background establish context, the description of the "medical data" remains too generic, referring to the Kaggle dataset without sufficient specificity. It is crucial to explicitly define the nature of the data, especially for readers unfamiliar with the dataset. Furthermore, Sections 3.2, 3.3, and 3.4 appear misplaced, and Section 3.5 is notably empty. The introduction effectively introduces the subject and clarifies the research motivation.

Experimental design

The methods lack critical details for replication, such as the train/test split and specific data types within the dataset. While data preprocessing is discussed, the inclusion of detailed formulas for standardization and normalization adds unnecessary complexity, as these are standard procedures that could be described more concisely. The evaluation methods, assessment metrics, and model selection are well-described, but presenting results for all six folds is redundant; reporting the mean and standard deviation would be more effective.

Validity of the findings

The research lacks an assessment of its impact and novelty and provides insufficient comparison with state-of-the-art work using the same dataset. The results are limited to a single dataset, restricting their general applicability. The conclusions, though well-stated, fail to adequately address the study's limitations, particularly the lack of comparison with other works and datasets. The authors should provide a more thorough evaluation of the research's context and indicate future directions, including addressing unresolved questions and limitations beyond those related to the dataset's differences from real-world clinical data.

---

## Round 0.2 · accepted · Accept

The reviewers assessed the new version of the manuscript and appreciated the changes made by the authors. Now I can recommend this article for acceptance and publication in the PeerJ Computer Science journal.

Reviewer 1 ·

Basic reporting

I found all responses from the authors satisfactory and agreed to accept the paper for publication. I don’t have further concerns over the article.

Experimental design

I found all responses from the authors satisfactory and agreed to accept the paper for publication. I don’t have further concerns over the article.

Validity of the findings

I found all responses from the authors satisfactory and agreed to accept the paper for publication. I don’t have further concerns over the article.

Additional comments

I found all responses from the authors satisfactory and agreed to accept the paper for publication. I don’t have further concerns over the article.

·

Basic reporting

All previously mentioned issues within basic reporting was addressed by the authors. Thank you for clarifying the questions within your manuscript.

Experimental design

The methodology and results sections now do not consist any longer of mutual and intertwined content.
Thank you for correcting the statements, as well as adding missing references, metrics, and description of how testing was conducted.

Validity of the findings

More information about experiments and setup is now given, metrics, values and results are listed appropriately. Therefore, the findings are assessed as scientifically valid.

Reviewer 3 ·

Basic reporting

The introduction effectively establishes the context and research motivation. The initial concerns regarding the generic description of the medical data have been thoroughly addressed. The detailed explanation of the Kaggle dataset, including the 44 clinical features (vital signs, lab results, and demographic information), provides crucial specificity for readers. Furthermore, the description of patient information as time-stamped events and the nature of the "SepsisLabel" as a target variable, allowing for predictive modeling of sepsis onset, significantly enhances clarity. The experimental sections (3.2, 3.3, 3.4) have been appropriately reorganized, and the formerly empty Section 3.5 has been rectified.

Experimental design

The revised manuscript now includes critical details necessary for replication, such as the train/test split and specific data types within the dataset. The authors have judiciously removed unnecessary formulas for standard preprocessing steps, improving conciseness. While the evaluation methods, assessment metrics, and model selection were already well-described, the consolidated presentation of the 6-fold cross-validation results in Table 3, reporting the mean and standard deviation for Accuracy, Precision, Recall, F1-Score, and AUC, is a significant improvement.

Validity of the findings

The authors have significantly enhanced the assessment of the research's impact and novelty by including a thorough comparison with state-of-the-art work using the same Kaggle dataset (Table 5). This comparison powerfully underscores the superior performance of the proposed methodology, which achieves 99.4% accuracy, far exceeding previous models. The detailed explanation of why their model excels, attributing it to specific architectural components and hyperparameter optimization via the African Vulture Optimization Algorithm (AVOA), effectively highlights its novelty.